solid-state physics

solar cells, radiation damage, electronic stopping power, semiconductor, interface

**Author for correspondence:**
Natalia E. Koval
e-mail: koval.tasha@gmail.com

# *Ab initio* electronic stopping power for protons in Ga$_{0.5}$In$_{0.5}$P/GaAs/Ge triple-junction solar cells for space applications

Natalia E. Koval[1], Fabiana Da Pieve[3]
and Emilio Artacho[1,2,4,5]

[1]CIC Nanogune BRTA, and [2]Donostia International Physics Center DIPC, 20018 Donostia-San Sebastián, Spain
[3]Royal Belgian Institute for Space Aeronomy BIRA-IASB, 1180 Brussels, Belgium
[4]Theory of Condensed Matter, Cavendish Laboratory, University of Cambridge, Cambridge CB3 0HE, UK
[5]Ikerbasque, Basque Foundation for Science, 48011 Bilbao, Spain

NEK, 0000-0002-9040-0512; EA, 0000-0001-9357-1547

Motivated by the radiation damage of solar panels in space, firstly, the results of Monte Carlo particle transport simulations are presented for proton impact on triple-junction Ga$_{0.5}$In$_{0.5}$P/ GaAs/Ge solar cells, showing the proton projectile penetration in the cells as a function of energy. It is followed by a systematic *ab initio* investigation of the electronic stopping power (ESP) for protons in different layers of the cell at the relevant velocities via real-time time-dependent density functional theory calculations. The ESP is found to depend significantly on different channelling conditions, which should affect the low-velocity damage predictions, and which are understood in terms of impact parameter and electron density along the path. Additionally, we explore the effect of the interface between the layers of the multilayer structure on the energy loss of a proton, along with the effect of strain in the lattice-matched solar cell. Both effects are found to be small compared with the main bulk effect. The interface energy loss has been found to increase with decreasing proton velocity, and in one case, there is an effective interface energy gain.

# 1. Introduction

With the new advancements in solar cell technology and the rapid expansion of space missions, understanding the key aspects

**Figure 1.** Schematic structure of the triple-junction solar cell. The active layers (GaAs and $Ga_{0.5}In_{0.5}P$) are lattice matched to the Ge substrate. The tunnelling junctions are usually made of p- and n-doped GaAs. Metallic contacts are often aluminium. A detailed structure can be found in [1].

which link the macroscopic response of solar cells of current and future spacecraft to the fundamental processes of particle stopping inside the target has acquired new relevance. Solar cells are key components of spacecraft and satellites as they provide power for navigation, communication, data handling, thermal control and functioning of the instrumentation. At present, the so-called triple-junction (TJ) solar cells, made of $Ga_{0.5}In_{0.5}P$, GaAs and Ge layers (see scheme in figure 1), are the state of the art for space applications, generally protected by an amorphous $SiO_2$ coverglass. All layers are usually lattice-matched to the substrate (Ge).

In space, radiation degrades the performance of the solar cells mainly by cumulative effects involving atomic displacements. Such displacements introduce consequent defect levels in the electronic structure, which in turn affect the output current. The atomic displacements are caused by the scattering of both primary incoming charges (protons) and secondary projectiles (displaced target atoms, protons, neutrons) generated by the primary radiation. For particle energies below 10–30 MeV (depending on the material), the damaging interaction is mainly due to the Coulomb repulsion between the proton and the target without affecting the nuclei, while hadronic interactions, involving subnuclear particles, take over at energies above 30–50 MeV. The Coulomb contribution to the non-ionizing energy loss (NIEL, the rate of energy deposition to atomic displacements, linked to the nuclear stopping) is the main one in the relevant ground testing energy range (0–10 MeV), using unidirectional fluxes on unshielded solar cells. Such energy range and configuration have been shown to be representative of the cumulative damage caused by omnidirectional space radiation on shielded solar cells, where energies in general go up to hundreds of MeV [2–4].

The (Coulomb) NIEL is often extracted from SRIM (the stopping and range of ions in matter) [5,6], from tabulated-data-based approaches like SR-NIEL [7], or from Monte Carlo particle transport codes such as Geant4 [8–10] (the latter two also provide the hadronic contribution). Despite some differences, some basic assumptions are similar in these approaches: crystalline order is not considered for the target (thus, there is no distinction between channelling and off-channelling conditions), cascades are treated within the binary-collision approximation (BCA), and the description of electronic stopping is based on the (perturbative) Bethe–Bloch theory at high energy and by the Lindhard dielectric theory combined with the local-density approximation (LDA) at lower energies ($E/n < 2$ MeV), implying that the electronic stopping power (ESP) at each point in the system is considered to be the same as that of a uniform electron gas of the same density. An *ad hoc* parametrization of Lindhard results is used at intermediate energies. In general, the low-energy physics in both SRIM and Monte Carlo particle transport codes, such as Geant4, strongly relies on phenomenology and the use of experimental data.

The influence of the ESP on defect production by irradiation of different ions at different energies is under intense investigation [11–24]. Recent studies, which address the ESP problem from the *ab initio* point of view and combine it with the molecular dynamics simulations of damage cascades, show the importance of including electronic stopping effects in cascade formation, in terms of the number of formed defects and cascade morphology [22–24]. Electronic excitation, as a result of the moving primary and secondary projectiles in the first stages of the radiation damage process, can significantly alter the subsequent ionic displacements and the number of defects formed in the solid. Thus, accurate description of the ESP processes is essential for the correct estimation of the radiation effect in materials.

Real-time time-dependent density-functional theory (RT-TDDFT) has opened the way for the predictive first-principles description of electronic stopping processes in condensed-matter systems, in

a nonlinear and non-perturbative way [25–31]. Recent RT-TDDFT studies on channelling conditions for proton impact have found a strong sensitivity of the ESP on crystal direction and impact parameter [32]. Deviations from velocity proportionality at very low velocity have also been obtained [32,33], consistent with the finite-velocity threshold observed in experiments [34]. Others have highlighted the different contributions of semicore electrons in channelling versus off-channelling conditions [35]. Local enhancement and reduction of the ESP for specific channels and velocity-dependent deviations from an ideal channelling trajectory were also reported [31], as well as the influence of a strongly anisotropic crystal structure on the stopping power [36].

The explicit consideration of channelling trajectories in the computer simulation of the radiation damage has been shown to be important [37]. Both experiments and simulations show that ion channelling plays a significant role in radiation damage cascades in crystalline materials. It has been observed that for an ion moving with energies in the keV range, the penetration depth is much larger due to the channelling effect, which affects the spatial distribution and the shape of the displacement cascade [38–40]. The channelling effect also contributes to the formation of the subcascades [40–42] and increases the time of the defect formation [43].

An initial Monte Carlo particle transport study (Geant4-based) shows that in a realistic space mission, the energy range of protons traversing all the three layers (deriving from both the primary space radiation and as products of nuclear fragmentation in the layers) extends until low energies. Low energy protons (up to few hundred keV), which stop inside the cell, have been shown to cause the most severe damage on TJ solar cells [44]. For all the reasons listed above, we chose the low-energy regime as the focus of this paper.

In this context, the aim of this work is to present a systematic investigation of the ESP of the sub-junction materials $Ga_{0.5}In_{0.5}P$, GaAs and Ge of space solar cells under low-energy (keV range) proton impact in channelling conditions, using RT-TDDFT as implemented in SIESTA [32,45]. We consider various channelling trajectories for the projectile (different impact parameters) inside the target. The results are compared with SRIM [5] data, which do not discriminate channelling trajectories from others.

Finally, an analysis is performed of the electronic energy loss at the interface between two upper layers of the TJ structure. To the best of our knowledge, no study has been presented investigating the change of the electronic response to particle radiation across the stack. The results presented here should contribute to understand to what extent channelling effects can be linked to both the electronic density and the atomic number of the element(s) in the compounds constituting the solar cell stack, which should serve as a basis to a further study of non-adiabatic MD simulations of cascades.

The key questions we ask in this work are: (i) how does low-energy channelling electronic stopping change with respect to the widely used random-trajectory average, as used in SRIM, (ii) how large can the differences among channels be, (iii) how different are the three materials, (iv) are there significant electronic effects at interfaces, and (v) what is the effect of the strain in the layers due to epitaxial growth?

# 2. Method

## 2.1. Monte Carlo particle transport simulations through the solar cell

One-dimensional Monte Carlo proton transport calculations of the isotropic protons through the multilayer stack of the TJ solar cell have been performed with the MUlti-LAyered Shielding SImulation Software (MULASSIS) [46] (based on Geant4 [8–10], available via the European Space Agency's (ESA) online interface SPENVIS [47]. Geant4 is a Monte Carlo particle transport simulation software used in application to high-energy, nuclear and accelerator physics, as well as studies in medical and space science. In order to simulate different energy ranges, different systems and configurations, a predefined set of physics models (the so-called physics lists) is included in the Geant4 toolkit, which allow for the description of hadronic and electromagnetic interactions with different accuracy. The hadron/ion interactions (in general for protons, neutrons and pions) at energies above 20 GeV, are described by a quark-gluon string (QGS) model [48]. At energies below 10 GeV, the binary intra-nuclear cascade (BIC) model for primary proton and neutron interactions with nuclei is used [48]. At low and intermediate energies, ionization and elastic scattering are the main processes of the interaction of protons with matter.

The energy loss due to electromagnetic interactions is calculated using the energy loss, range and inverse range tables pre-computed at initialization of Geant4 for each material [49,50]. In all Geant4 electromagnetic physics configurations, two models are used for proton ionization [51]: the PSTAR/

SRIM stopping power [5,52] for proton energies below 2 MeV, and the Bethe–Bloch formula with shell, Barkas and Bloch, and density effect corrections [53] for energies above 2 MeV. The G4EmStandardPhysics_option3 was used for the simulation of electromagnetic interactions. By choosing this physics list, the appropriate descriptions of the electromagnetic processes are automatically applied for different particles in different energy ranges.

A simplified solar cell structure with a typical $SiO_2$ amorphous 100 μm thick coverglass [54], and with the $Ga_{0.5}In_{0.5}P$, GaAs and Ge layers having 0.8, 2 and 175 μm thickness, respectively, is used, similarly to [55]. The omnidirectional fluxes of trapped protons, covering the energy range of 0.1–400 MeV (Earth's radiation belt, on the basis of data from more than 20 satellites), have been generated via the NASA AP-8 model [56].

The transport of ions in matter (TRIM) part of the SRIM code is used to calculate the proton track trajectories through the solar cell stack. TRIM is a Monte Carlo code that calculates the interactions of energetic ions with amorphous targets using several approximations, such as binary collisions only, an analytical formula for the ion–ion interactions, and the so-called concept of a free-flight-path between the collisions, such that only significant collisions are evaluated [5,6].

## 2.2. Electronic stopping power from real-time time-dependent density-functional theory

The ground state configuration of each system is calculated using the DFT implementation of the SIESTA code (see appendix A for details) [45] with the projectile placed at the initial position for each trajectory (specified in §3.2).

The ground state Kohn–Sham orbitals of the system are obtained by solving the Kohn–Sham equations self-consistently until the total ground state energy is converged. The exchange-correlation functional uses the local-density approximation (LDA) in the Ceperley–Alder form [57]. Norm-conserving Troullier–Martins [58] relativistic pseudopotentials are used to replace the core electrons (see appendix A for the parameters used to generate pseudopotentials). It is known that core electrons can have large effects on the ESP (e.g. [59]). However, here we only test the effect of core electrons for one proton trajectory in Ge, for all the others, we consider the interaction of the projectile with valence electrons, given the low charge of the projectile, and the relatively low-velocity range. In such limit, the effect of core electrons is negligible as we demonstrate in our work (see also the validation with experiments of similar calculations for protons in Ge in [32]). The valence electrons are represented by a double-$\zeta$ polarized basis set of numerical atomic orbitals defined as specified in the appendix A.

We use real-time time-dependent DFT (RT-TDDFT) implementation of the SIESTA method [32,60] to evolve the electronic orbitals in time. The time-dependent Kohn–Sham equations are solved by real-time propagation for discretized time, using the Crank–Nicholson scheme [60,61] with a time step $dt = 1$ attosecond. The effect of the moving basis set is accounted for by a Löwdin transformation, as described in [62,63], which is known to be adequate for relatively low velocity of the moving basis orbitals, and offers strictly unitary propagation for finite $dt$.

The focus of this work is on the purely ESP for constant velocity projectiles. The forces on the nuclei of the target atoms and on the projectile itself are therefore disregarded for the time propagation, thereby describing nuclear dynamics with frozen host nuclei and a constant velocity projectile, as done in many similar studies [25,29,32,59]. It allows the clean separation of the electronic and nuclear contributions to the total stopping. Fixing the atomic positions has a negligible effect for the simulations performed in this work, given the fact that the projectile traverses the simulation box in about $t \leq 4$ fs, and the ionic displacements on that time scale are negligible (head-on collisions are avoided in channelling conditions).

All three target materials have a diamond cubic structure with a similar value of the lattice constant. We used the theoretical lattice constant of 5.59 Å for Ge (for which the experimental value is 5.66 Å) in order to compare our results with Ullah *et al.* [32]. For other layers, we used experimental lattice constants of 5.65 Å for GaAs, and 5.66 Å for $Ga_{0.5}In_{0.5}P$ (average between the values for GaP, 5.45 Å, and InP, 5.87 Å). The change in the lattice constant by 1.7% only changes the stopping power by a negligible 0.28% according to our results (see §3.6).

The targets are modelled by a super-cell of 96 atoms, constructed by multiplying the conventional unit cell (which consists of 8 atoms) by $2 \times 2 \times 3$ in the $x, y$ and $z$ directions, respectively. A $2 \times 2 \times 1$ Monkhorst–Pack [64] **k**-point mesh is used, which corresponds to a **k**-grid inverse cut-off of 8.4 Å. The **k**-points are displaced to the centres of the grid cells. Two channels in the target super-cell are chosen as trajectories for the proton, along the [001] and [011] crystallographic directions, as shown in figure 2 for the case of Ge. Besides the channelling trajectories, and defining the channel centre as the axis furthest away from all atoms, we also choose various off-centre parallel channel trajectories with different impact

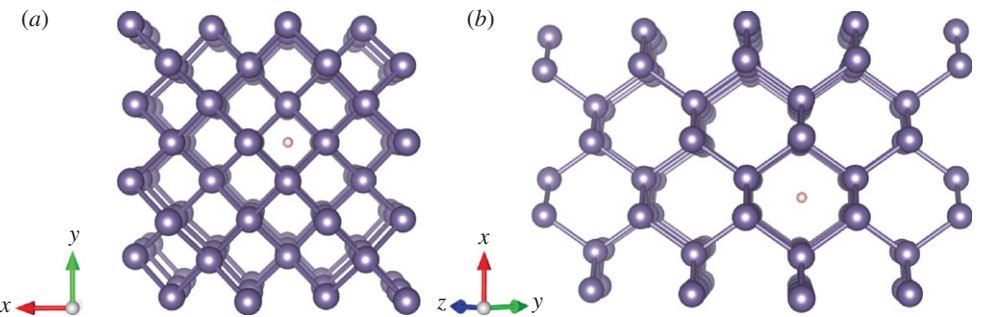

**Figure 2.** Ge super-cell (96 atoms). A proton position is shown in the centre of (*a*) [001] channel and (*b*) [011] channel. GaAs and $Ga_{0.5}In_{0.5}P$ have a similar structure.

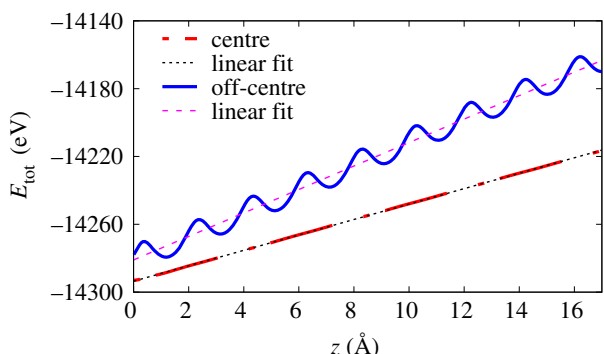

**Figure 3.** Position-dependent total electronic energy for two trajectories of the proton moving through Ge with velocity *v* = 0.5 a.u. along the [011] direction. The energy is shown for the proton moving along the centre of the channel (impact parameter of 1.98 Å, red long-dashed line) and along the off-centre channelling trajectory (impact parameter of 0.7 Å, blue line). The linear fitting is shown for each case.

parameters, i.e. at different closest distances of the proton from the host atoms. The same super-cell is used in simulations for [001] and [011] channels, since the length of the trajectory is similar in both directions.

The ESP is calculated as $S = dE_{tot}/dz$, the derivative of the Kohn–Sham total electronic energy with respect to projectile position *z* along the constant-velocity path [25]. $E_{tot}(t)$ is taken as approximate density-functional for $\langle H \rangle(t)$. The average of *S* is obtained from the slope of a linear fitting to $E_{tot}(z)$ as shown in figure 3. Figure 3 shows the energy for the case of a proton moving in Ge with a velocity of 0.5 a.u. along the centre of the [011] channel and for an off-centre trajectory along the same direction. The oscillations in the figure reflect the periodicity of the crystal. They are quite prominent for the low-impact factor case, but are not noticeable for the mid-channel trajectory, along which the density is quite homogeneous.

All the results in the following are compared with the SRIM [5] data for the ESP. SRIM results are obtained semi-empirically by averaging over a number of different incident directions with distinct impact parameters, with no explicit consideration for the channelling conditions studied in our work. The density of the target materials that we used in the SRIM calculations were computed from the unit-cell volume obtained in the DFT ground state calculations.

# 3. Results and discussion

## 3.1. Monte Carlo particle transport simulations

Figure 4 depicts the results of the 1D-Monte Carlo (Geant4-based) simulations of the propagation of the isotropic protons through the solar cell stack. The simulations are performed for the representative example case of the protons trapped in the Van Allen belts accumulated during a 3-year International Space Station-like mission. The first step is the calculation of the slowed-down proton differential spectrum through the slab of the protective coverglass $SiO_2$ layer. The results in figure 4 clearly show that the coverglass lowers the fluences of lower energy particles, leaving almost unaffected the high-energy portion of the proton spectrum. Those protons are not only the primary impacting protons but also those generated by the nuclear interactions in the target. Overall, the results show that, for a realistic scenario,

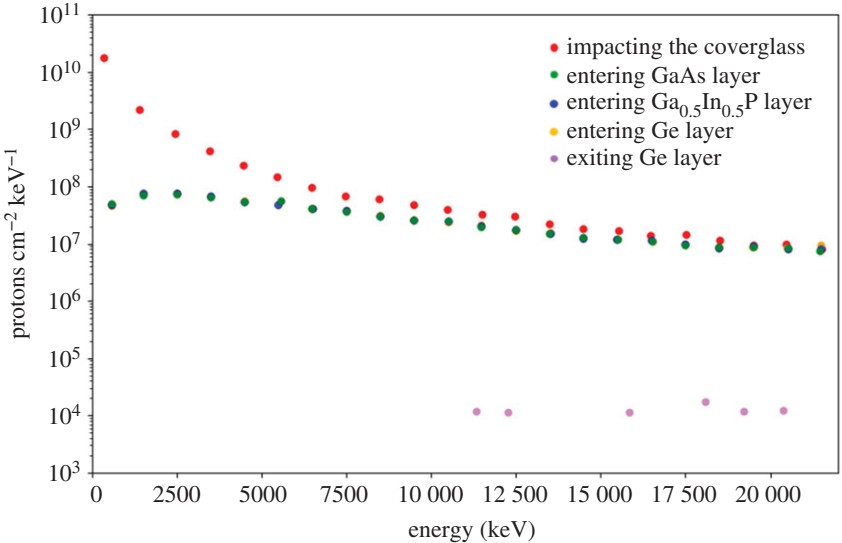

**Figure 4.** Fluence of the omnidirectional primary proton radiation across all layers of the solar cell structure (coverglass included).

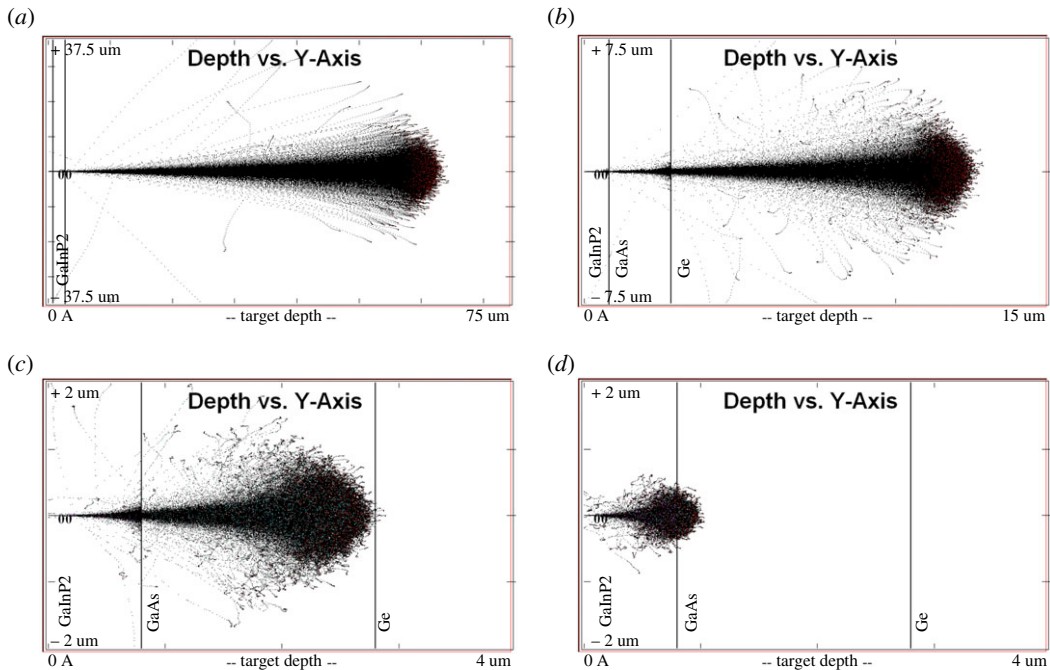

**Figure 5.** Ion track trajectory output from TRIM for unidirectional monoenergetic protons. The thickness of the layers (density) is 0.8 μm (4.21195 g cm$^{-3}$) for Ga$_{0.5}$In$_{0.5}$P, 2 μm (5.8155 g cm$^{-3}$) for GaAs and 175 μm (5.35 g cm$^{-3}$) for Ge. The energies of the impacting protons are representative of the low energy ranges used by recent studies: (*a*) 3 MeV [55], (*b*) 1 MeV [65–67], (*c*) 0.3 MeV [68] and (*d*) 0.1 MeV [69]. Note different scale on both axes in different panels.

once the particles are slowed down by the 100 μm SiO$_2$ coverglass, the energies of the particles entering all the other layers change to a negligible amount (except for the particles exiting the bottom of the bottom Ge subcell, whose spectrum is considerably 'hardened', i.e. energies lower than approximately 12 MeV do not exit the whole stack (figure 4). Within the protective amorphous oxide cap layer, ions will be scattered with no channelling, and only a few will find the channels. Contrarily, in the crystalline materials constituting the other layers, channelling conditions may either enhance the depth of penetration or exhibit, for specific impact parameters, blocking trajectories. The fluence always contains particles extending to the low-energy regime of one-hundredth to tenths of keV, for essentially all layers in the structure.

In figure 5, we report the results of the SRIM (TRIM) Monte Carlo simulations, which are performed, as it is the case in ground testing studies, assuming unshielded configurations and unidirectional (normal

incidence), monoenergetic proton irradiation. TRIM only deals with Coulomb scattering. Thus, the calculated damage is that produced by the projectile itself, the primary knock-on atoms (PKAs), and consequently generated secondary (SKAs), and not by eventual 'secondary particles' produced in nuclear reactions. The energies chosen for the impacting protons are representative of the low energy ranges used by recent studies (3 MeV [55], 1 MeV [65–67], 0.3 MeV [68] and 0.1 MeV [69]). It is important to note that, for such impacting energies, the particles will stop inside the solar cell, in the Ge layer (figure 5a,b), in the GaAs layer (figure 5c), or between the two active layers $Ga_{0.5}In_{0.5}P/GaAs$ (figure 5d). Thus, the relevant range of impact energies that is necessary to consider in the ESP calculations extends down to 0 keV. Since the damage produced per unit path length increases as the proton energy decreases and some protons stop within the active region of the cell (figure 5), a non-uniform damage can be induced at such low energies. Thus, the application of the current modelling approach, the so-called DDD (displacement damage dose) model exhibits some limitations for a relevant energy range (1–10 MeV), since it is based on a uniformity of the damage [70] and on an observed linear dependence between the NIEL (i.e. the energy deposited to the atomic displacements) and the damage (i.e. the number of defects produced). Thus the NIEL concept should be improved in order to take into account the details of the structure of the system and the dependence of the NIEL on depth (instead of the NIEL based on the incident energy [69]), which can change in channelling versus non-channelling conditions). If an ion enters a crystalline target along a direction with a low Miller index, the energy-dependent ratio of electronic and nuclear stopping, which is valid for an amorphous material, will no longer be valid.

## 3.2. Electronic stopping power in channelling conditions, comparison to SRIM

The ESP as a function of the proton velocity is shown in figure 6 for three layers of the TJ solar cell and for two channels: [001] and [011]. The corresponding incident energies of the proton can be obtained from $E = m_p v^2/2$, where $m_p = 1836$ a.u. is the proton mass (0.999–20.234 keV for the chosen range of velocities of 0.2–0.9 a.u.). The trajectories of the proton in the [001] and [011] channels are schematically depicted in the insets of figure 6. Trajectory 1 corresponds to the centre of the channel in all the cases. The impact parameter for the off-centre trajectories closest to the rows of atoms (and to the axis between two atoms in the case of the channel [011]) is 0.7 Å (approximately half the distance from the centre of the [001] channel to the target atom). In Ge, we also consider additional trajectories with the impact parameter equal to 1.05 Å for trajectory 2 in figure 6a and 1.4 and 1.86 Å for trajectories 2 and 4, respectively, in figure 6b. The orange curve in each panel of figure 6 shows the ESP calculated with SRIM [5] using the following values of the density: 5.53 g cm$^{-3}$ for Ge, 5.32 g cm$^{-3}$ for GaAs, and 5.52 g cm$^{-3}$ for $Ga_{0.5}In_{0.5}P$, which correspond to the lattice constants used in RT-TDDFT calculations.

In Ge, the values of the ESP are very similar for all three [001] trajectories (figure 6a), since the electronic density distribution does not vary much inside this narrow channel. This can be seen in figure 7 in which we plot the electron density averaged along the [001] channel as a function of position in the perpendicular plane for all three layers. The average is obtained from the density calculated for 160 [x, y] planes along the z-axis with a step of dz = 0.1 Å. Figure 7c shows that the average density is much more homogeneous throughout the unit cell in Ge as compared with the other two layers (figure 7b,c) and that the density varies only slightly along three trajectories.

A larger dispersion of ESP is observed in the [011] channel in Ge (figure 6b), being a consequence of a larger variation of the electron density inside this channel [32]. The lowest ESP comes from the mid-channel, as it can be reasonably expected given the lower electron density encountered in such trajectory. For a slow proton moving along the [011] channel, the stopping power is similar for all the trajectories. However, at higher velocities, the stopping power depends on the impact parameter quite significantly. As it has been noted in [32], this behaviour is a consequence of the strong correlation between the stopping power and the average local density within a small radius of the impact parameter, and that this radius is larger for a slower projectile.

On average, our results for the ESP in Ge are slightly lower than the SRIM for the proton moving in both channels, except for the trajectories 3 and 5 in the [011] channel, for which the ESP approaches the SRIM values. This suggests that in the SRIM calculations, the average density is higher than the density on most trajectories in the RT-TDDFT calculations for Ge. The average valence density in Ge is about 0.179 electrons Å$^{-3}$, which corresponds to the trajectory 3 in figure 7c, while the average density along the trajectories 1 and 2 is slightly lower. Present results for a proton moving on the mid-channel trajectories along the directions [001] and [011] in Ge are in agreement with the results previously obtained by Ullah et al. [32].

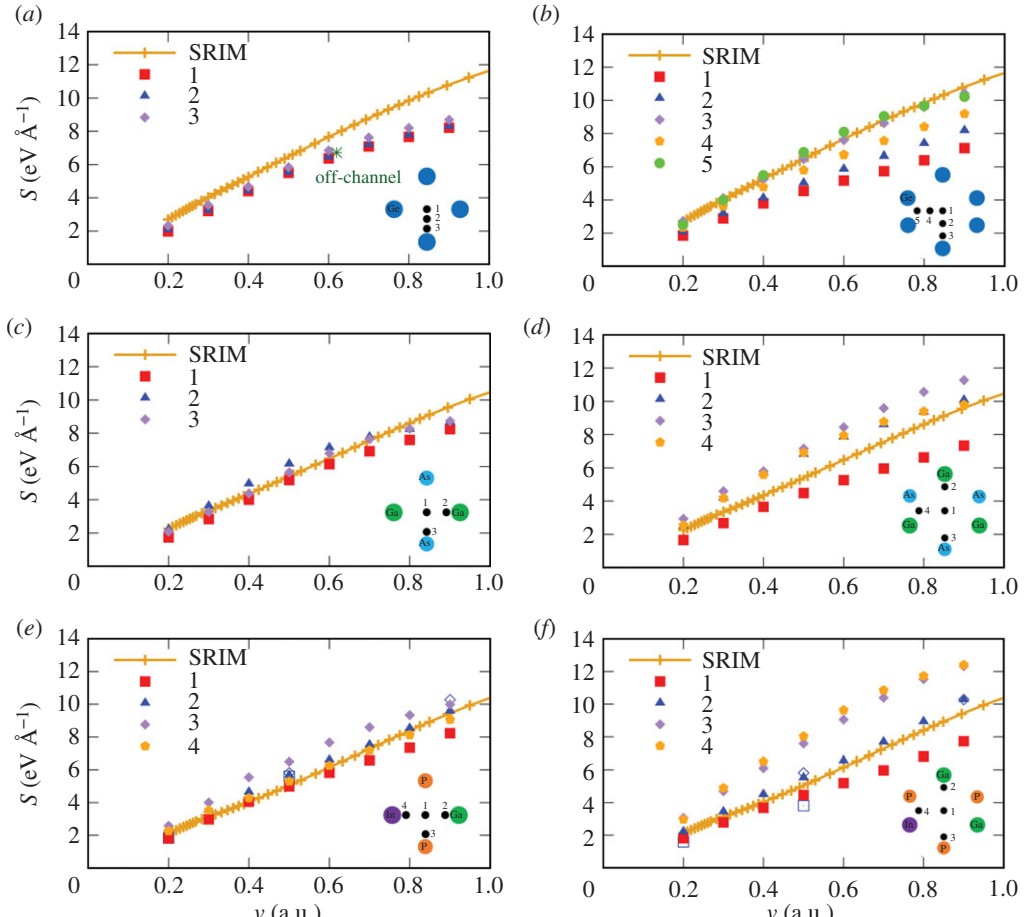

**Figure 6.** Electronic stopping power for a proton in $Ga_{0.5}In_{0.5}P$/GaAs/Ge compared to SRIM. The stopping power is shown as a function of velocity for the channel [001] in (*a*) Ge, (*c*) GaAs and (*e*) $Ga_{0.5}In_{0.5}P$; and for the channel [011] in (*b*) Ge, (*d*) GaAs and (*f*) $Ga_{0.5}In_{0.5}P$. SRIM results are shown as the solid curve with crosses for each layer. A schematic of the trajectories of the proton in each channel are shown in the insets. Green asterisk in (*a*) shows our result for an off-channel trajectory (see text for details). Empty symbols in (*e*) and (*f*) are the results from Lee & Schleife [31], the squares and diamonds correspond to the channelling and off-channelling results, respectively (see text for details).

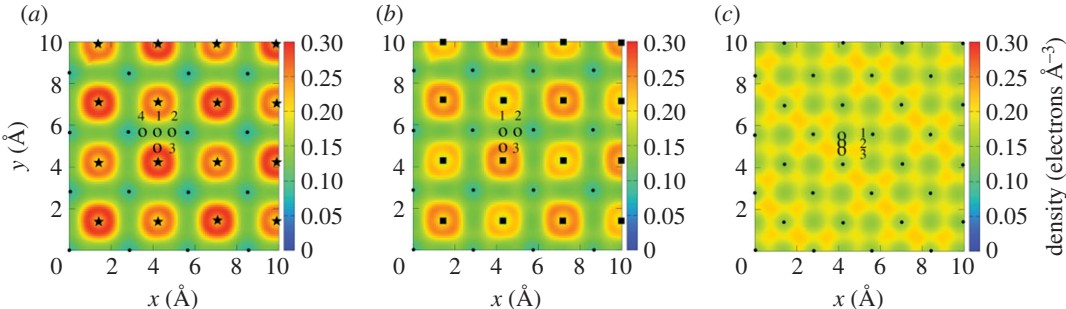

**Figure 7.** Electron density averaged along the [001] channel as a function of position in the perpendicular plane for $Ga_{0.5}In_{0.5}P$, GaAs and Ge. The trajectories of the proton in all systems are indicated with open circles and are the same as in the insets of figure 6. (*a*) Average density in $Ga_{0.5}In_{0.5}P$, black dots indicate positions of the alternating Ga/In atoms, while black stars indicate positions of the P atoms; (*b*) average density in GaAs, black dots and squares show positions of the Ga and As atoms, respectively; (*c*) average density in Ge, black dots show positions of the Ge atoms. The colour scale is the same for all three panels.

In figure 6*a*, we show as well the ESP for an off-channel proton trajectory (random direction, not parallel to any channel) in order to see how well it compares to SRIM. The random trajectory is arbitrarily chosen so that it makes 14° with the direction [001] (*z*-axis). The initial projectile position is

at the centre of the [001] channel at a distance 0 Å from the first atomic plane perpendicular to this direction. The velocity of the proton is 0.62 a.u. (0.6 a.u. along the z-axis and 0.15 a.u. along the x-axis). The calculation was performed during 6 fs, four times the time needed to cross the simulation box at this velocity. Thus it is equivalent to four different random trajectories [31]. The resulting ESP for the off-channel trajectory is in between the values for two off-centre channelling trajectories 2 and 3, and thus does not bring the channelling results closer to SRIM. It is known that longer trajectories would affect our results, but on a smaller scale than the observed discrepancy [31]. A possible reason for that is the absence of core electrons, as discussed in the next subsection. However, SRIM is expected to be inaccurate on that scale as well [71], and further expense in calculations is therefore not justified.

In GaAs, similarly to Ge, the ESP for the proton in the [001] channel is almost independent on the impact parameter and agrees perfectly with the SRIM data (figure 6c). Only the ESP for the proton on trajectory 3 is slightly higher than on 1 and 2, given the higher average electron density on this trajectory (figure 7b). In the [011] channel, the ESP is below SRIM for the proton moving on the mid-channel trajectory, while it is above SRIM for all the off-centre trajectories with small impact parameter (i.e. higher electron density) (figure 6d). At proton velocities below 0.5 a.u., the ESP in the [011] channel is similar for all the off-centre trajectories. At higher velocities, however, the ESP corresponding to the trajectory 3, closest to the As atoms, is slightly higher. This could indicate that the maximum of the ESP is located at different velocities for different proton trajectories.

In the case of the $Ga_{0.5}In_{0.5}P$ layer, again, on average, our results are in good agreement with SRIM (figure 6e,f). However, the dependence of the ESP on the trajectory is observed in both channels. For the proton moving on trajectories close to Ga and In atoms in both channels, the ESP values are very close, owing to the fact that both elements have three electrons in the outer shell. A much higher ESP is obtained for the trajectories close to P, which have five valence electrons, and thus, the electron density is higher along these trajectories (trajectory 3 in figure 7a). Analysis of the average electron density for different proton trajectories in the direction [001] in $Ga_{0.5}In_{0.5}P$, indicated with open circles in figure 7a, shows that the density is similar for trajectories 1, 2 and 4, but is higher by a factor of 2 along the trajectory 3. However, the stopping power for a proton moving on the trajectory 4 is only 1.4 (maximum, at $v = 0.7$ a.u.) times larger than the one on the trajectory 1 (figure 6e). Thus, the stopping power is not linear with the average density along the proton path.

In figure 6e,f, we also compare our results to the calculations by Lee & Schleife [31]. Similarly to our work, the ESP is obtained with RT-TDDFT (LDA), but using a plane-wave basis set. They are represented as empty squares for the stopping along the mid-channel trajectory, and as empty diamonds for the off-channel trajectory (i.e. random trajectory through the lattice, not parallel to any of the main crystallographic directions). There are small deviations between the results of both approaches, probably due to basis sets and pseudopotentials, but the main results, trends and conclusions are the same. Those deviations are smaller than the spread for different trajectories, and the differences with respect to SRIM.

Small deviation of the ESP from the SRIM behaviour observed at high velocities (especially in Ge) is due to two reasons. First, closer to the maximum of the stopping power, the effect of the inner electrons starts to be important, which is not taken into account in these calculations. We will discuss this effect in the next subsection. Another reason is that the Sankey integrator used in SIESTA to propagate the KS states in time [63] is known to be problematic at higher energies.

## 3.3. Comparative analysis of the electronic stopping power in different layers

We further analyse our results by comparing the values of the ESP for equivalent proton trajectories inside different layers of the TJ solar cell as shown in figure 8.

Overall, the ESP do not vary much from material to material for the proton moving on the mid-centre trajectories (figure 8a,b). For the smallest impact parameter of 0.7 Å in the [001] channel (figure 8c), we show the ESP for proton trajectories close to different elements in each material. The highest ESP corresponds to the trajectory close to P atoms in $Ga_{0.5}In_{0.5}P$. Figure 8d shows the ESP for the proton trajectories 5 (Ge) and 4 (GaAs and $Ga_{0.5}In_{0.5}P$) in the [011] channel (see trajectories in figure 6). Here as well, the largest ESP is obtained for the proton on the trajectory close to P atoms.

The importance of core electrons was checked for the case of Ge (figure 8a) by including 3d electrons into the valence shell (see appendix for details of the calculations). At velocities below 0.6 a.u., the addition of core electrons has no effect on the ESP. However, at velocity 0.8 a.u, the ESP is notably higher when 3d electrons are included in the valence, for this particular trajectory.

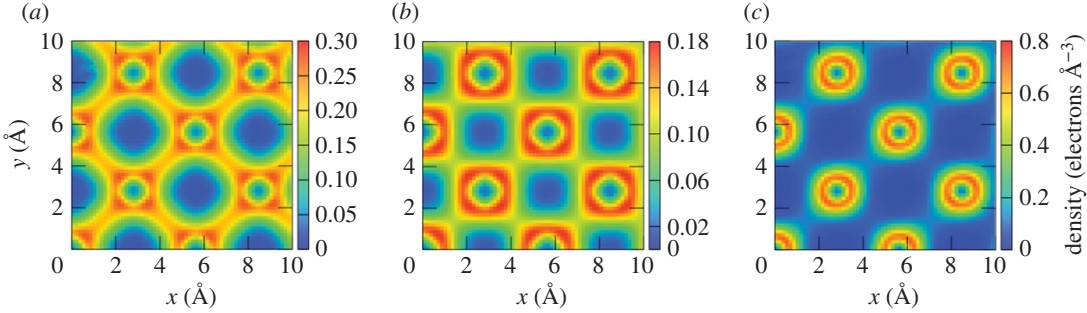

**Figure 8.** Electronic stopping power for a proton in $Ga_{0.5}In_{0.5}P/GaAs/Ge$, a comparison of different materials. The stopping power is shown as a function of velocity for (*a*) direction [001], centre of the channel. Red diamonds show the results for Ge including semicore electrons; (*b*) direction [011], centre of the channel; (*c*) direction [001], impact parameter of 0.7 Å; (*d*) direction [011], impact parameter of 0.7 Å from the edge of the hexagon. The inset on each panel shows a schematic view of the channel section and the proton position according to the impact parameter.

**Figure 9.** Electron density for three planes in $Ga_{0.5}In_{0.5}P$ perpendicular to the direction [001]. The planes are crossing through (*a*) Ga; (*b*) In; (*c*) P atoms.

Since $Ga_{0.5}In_{0.5}P$ is a compound with the largest number of constituent species in TJ solar cells, it is interesting to look at the electron density distribution inside it and try to correlate it with the observed ESP. Figure 9 shows the electron density for three different planes perpendicular to the [001] direction. Each plane crosses the positions of atoms of only one type (Ga, In or P). From figure 9, it is obvious that the electron density in the vicinity of the P atoms is much higher, which explains the largest ESP for protons on the trajectories 3 and 4 (figure 6).

Not only the value, but the spatial distribution of the electron density varies in different planes as well. Figure 10 shows the density as a function of $y$-coordinate for all three panels of figure 9 corresponding to $x = 0$. Thus three curves in figure 10 show the density distribution around one single atom of Ga, In and P, respectively. The system of coordinates is chosen so that all the atoms are centred at the origin ($y = 0$). The density, going down towards the nuclear position, reflects the absence of core electrons in the calculations. The maximum density around P atom is higher and is closer to the origin. Thus the projectile moving at 0.7 Å from the atomic position (indicated by the vertical dashed line in figure 10), in the case of P, is moving in the area of much higher electron density than in the case of Ga and In.

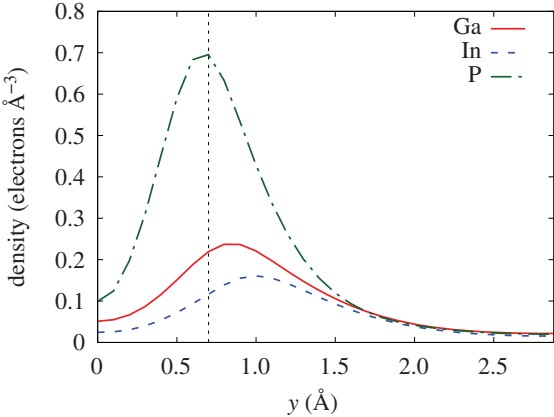

**Figure 10.** Electron density along the $y$-axis, corresponding to the cut through $x = 0$ in three panels of figure 9. The vertical dashed line shows the position of the proton in the off-centre channelling trajectory at 0.7 Å from the atomic position, corresponding to trajectories 2, 3 and 4 in the inset of figure 6$e$.

## 3.4. Comparison with the homogeneous electron gas model

In order to show the nonlinearity of the ESP obtained with RT-TDDFT, we compare our results with the homogeneous electron gas (HEG) approximation. In HEG, the electron density is defined through the so-called Wigner–Seitz radius $r_s$, a one-electron radius, as $n = 3/(4\pi r_s^3)$. The stopping power in HEG is calculated as a product of the friction coefficient and proton velocity. The friction coefficients corresponding to different $r_s$ are taken from [72]. In figure 11, we show the stopping power as a function of the average electron density in $Ga_{0.5}In_{0.5}P$ channel [001]. The lower density points (three points close to each other) correspond to the trajectories 1, 2 and 3, and the higher density corresponds to the trajectory 4. The crosses show the results of the HEG approximation, while the squares show the RT-TDDFT stopping power. The comparison is shown for three different velocities of the proton. At lower velocities, the agreement is very good. At higher velocity, however, the importance of the nonlinear effects is clearly observed as the HEG results overestimate our results.

## 3.5. Interface effects on the proton energy loss

In order to study the effect of an interface on the energy loss of the proton, we constructed a super-cell of the interface between the two upper layers of the TJ solar cell, i.e. $Ga_{0.5}In_{0.5}P$/GaAs along the crystallographic direction [001] of the epitaxial growth of the lattice-matched solar cells (figure 12$a$). The super-cell consists of 192 atoms and has the lattice parameter average between the pure bulk $Ga_{0.5}In_{0.5}P$ and GaAs. We have chosen two trajectories of the proton through the interface, the channelling one (figure 12$b$) and the off-centre channelling trajectory (figure 12$c$), in which the proton first passes close to P and then to As atomic rows in the direction [001] with an impact parameter of 0.7 Å.

The difference in the energy loss of the proton moving through the interface and through the pure $Ga_{0.5}In_{0.5}P$ and GaAs super-cells of the same size is shown in figure 13 for velocities 0.3 a.u. and 0.6 a.u. for both trajectories. The interface between two materials is located at $z_{int} = 16.26$ Å. The proton first moves though the $Ga_{0.5}In_{0.5}P$, hence the difference in the proton energy loss $\Delta(dE_{tot}(z)) = dE_{int}(z) - dE_{GaInP_2}(z)$ is equal to zero until $z \simeq z_{int}$ in all the cases. For a similar reason, starting from $z \simeq z_{int}$, the value of $\Delta(dE_{tot}(z)) = dE_{int}(z) - dE_{GaAs}(z)$ is constant. Interestingly, the slope of $\Delta(dE_{tot}(z))$ has opposite trends for the two trajectories. In the case of the mid-channel trajectory (figure 13$a,c$), the proton energy loss in GaAs is larger than in $Ga_{0.5}In_{0.5}P$. On the off-centre trajectory (figure 13$b,d$), however, the behaviour is reversed. This change of slope occurs due to the fact that the electron density around P atoms is more localized (figure 9), and is slightly lower than around Ga in the centre of the [001] channel ($y = 2.1$ Å, figure 10). Thus, the electron density in the centre of the channel in $Ga_{0.5}In_{0.5}P$ is expected to be lower than in GaAs, giving rise to lower energy loss of the proton moving on this trajectory in $Ga_{0.5}In_{0.5}P$. On the off-centre trajectory, on the contrary, the density around P is much higher. This behaviour is, however, a bulk effect not related to the interface itself.

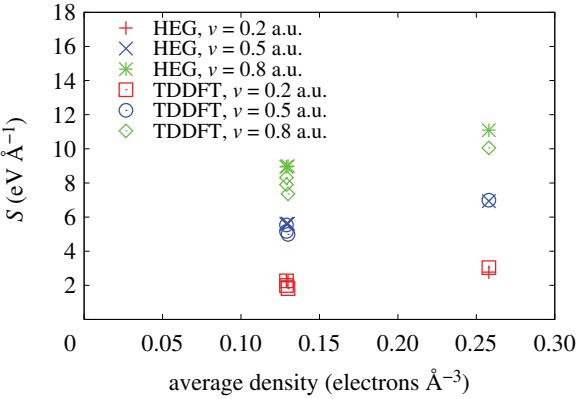

**Figure 11.** Electronic stopping power from RT-TDDFT compared to the HEG approximation. The stopping power as a function of the average electron density in $Ga_{0.5}In_{0.5}P$ channel [001] from RT-TDDFT for velocities of the proton $v = 0.2$ a.u. (open squares), $v = 0.5$ a.u. (open circles) and $v = 0.8$ a.u. (open diamonds); and HEG approximation for velocities $v = 0.2$ a.u. (pluses), $v = 0.5$ a.u. (crosses) and $v = 0.8$ a.u. (asterisks).

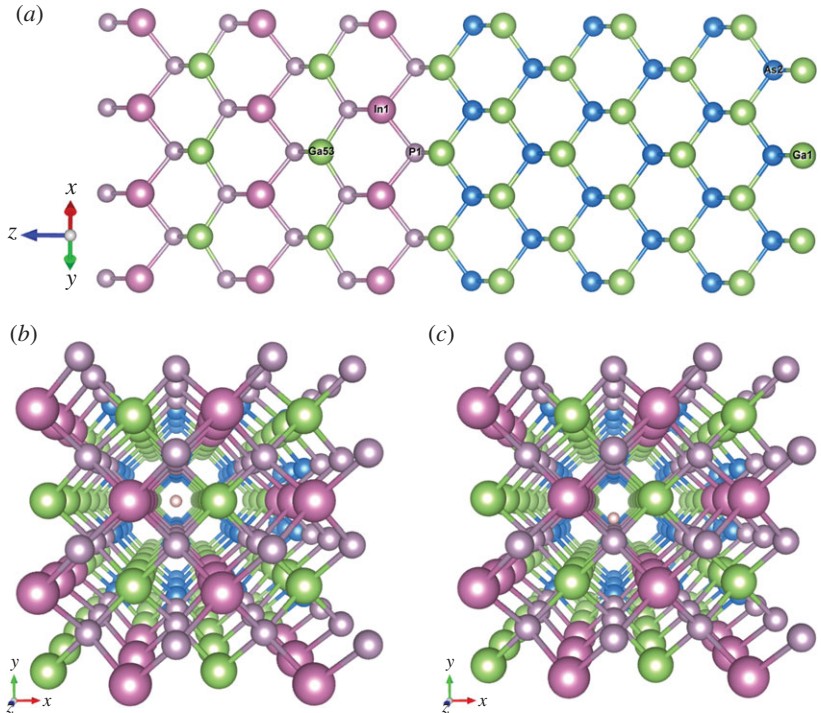

**Figure 12.** Atomic structure of the interface. (a) Structure of the interface between $Ga_{0.5}In_{0.5}P$ and GaAs, (b) proton in the centre of the channel [001], (c) proton in the off-centre channel at 0.7 Å from P and As atoms of the interface structure.

To quantify the interface effect and separate it from the bulk effects, we have calculated the proton energy loss on a given part of the trajectory equal to the length of two unit cells on each side of the interface structure ($dE_{int(Ga_{0.5}In_{0.5}P)}$ and $dE_{int(GaAs)}$) and for the equivalent part of the trajectory of pure structures ($dE_{Ga_{0.5}In_{0.5}P}$ and $dE_{GaAs}$). The difference between the energy loss of a proton moving through the interface structure and the sum of the energy losses of a proton moving through the corresponding intervals of pure structures, gives us the difference due to the interface: $\Delta E_{int} = dE_{int(Ga_{0.5}In_{0.5}P)} + dE_{int(GaAs)} - dE_{Ga_{0.5}In_{0.5}P} - dE_{GaAs}$. The values we obtained for $\Delta E_{int}$ for centre and off-centre channelling trajectories are 0.39 and 0.60 eV for velocity 0.3 a.u., and $-0.075$ and 0.072 eV for velocity 0.6 a.u., respectively. Thus, the interface effect is higher at lower velocities. The energy loss due to the interface even becomes negative at high velocity in the case of centre-channel trajectory, meaning effectively an energy gain. These differences, however, are 1–2 orders of magnitude smaller than the difference between the materials, i.e. the bulk effect.

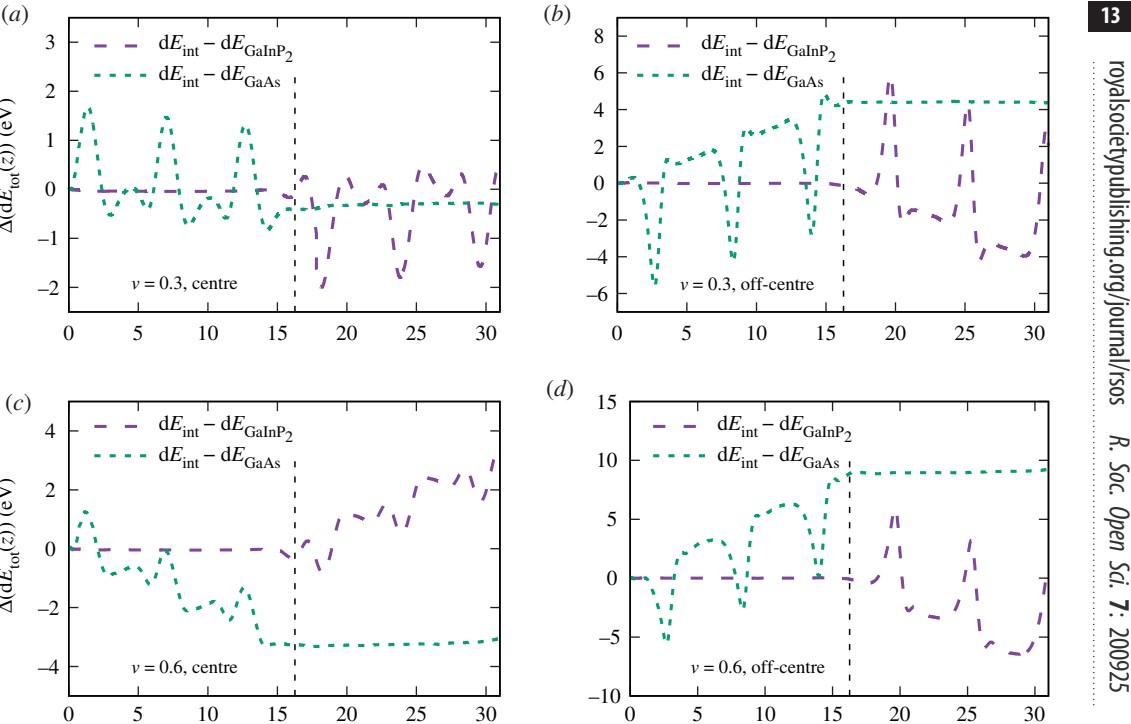

**Figure 13.** Difference in the proton energy loss between pure GaAs and $Ga_{0.5}In_{0.5}P$ and the interface between the two materials. The quantity plotted is $\Delta(dE_{tot}(z)) = dE_{int}(z) - dE_{pure}(z)$, where $dE(z) = E(z) - E(z=0)$ for (a) channelling trajectory, $v = 0.3$ a.u.; (b) impact parameter 0.7 Å, $v = 0.3$ a.u.; (c) channelling trajectory, $v = 0.6$ a.u.; (d) impact parameter 0.7 Å, $v = 0.6$ a.u., as indicated in each panel. The interface between GaAs and $Ga_{0.5}In_{0.5}P$ is located at 16.26 Å and is shown by the vertical dashed line in each panel.

## 3.6. Effect of strain in lattice-matched solar cells on the electronic stopping power

Most of the TJ solar cells currently used for space applications are lattice-matched. Ge is the thickest layer used as a substrate. The GaAs layer is grown on top of Ge and the $Ga_{0.5}In_{0.5}P$ is grown on top of the GaAs layer, with the lattices of different layers matching at the interface. The experimental lattice constants of these materials are very similar, 5.658 Å for Ge, 5.653 Å for GaAs (only 0.08% different from Ge) and 5.6596 Å for $Ga_{0.5}In_{0.5}P$. This small difference in the lattice constant leads to a strain in the GaAs and $Ga_{0.5}In_{0.5}P$ layers. We have analysed the effect of strain in two cases. In the first case, we have calculated the stopping power for a proton moving with $v = 0.8$ a.u. in GaAs with lattice constants 5.65 and 5.75 Å (the difference of 1.74%). The stopping power only changed by 0.28%. In another case, we compared the stopping power for a proton moving with $v = 0.5$ a.u. in GaAs with lattice constants 5.75 and 5.57 Å (3.13% different). The difference in the stopping power was only 1.13%. The actual differences in the lattice constants of the three layers of TJ solar cell are much smaller, and the changes in the stopping power can be considered negligible.

## 4. Conclusion

In this work, using RT-TDDFT, we have calculated the ESP of the three sub-junctions of the TJ solar cells for the impacting protons. Kinetic energies of the proton are considered in the keV range, which we have shown to be relevant in the transport of protons through the full stack in a realistic scenario. We compared our results for channelling conditions to the SRIM semi-empirical stopping power. Our results have shown that in Ge, the stopping power in the channel [001] is almost independent of the trajectory, while in both compounds (GaAs and $Ga_{0.5}In_{0.5}P$) the electronic stopping varies more along different trajectories, affected by varying electron density. In the case of the channel [011], strong dependence of the stopping power on the trajectory is observed in all three materials. The effect is more significant at higher proton velocities.

The change of sign for the slope of the energy loss difference between GaAs and $Ga_{0.5}In_{0.5}P$ in the interface structure is a bulk effect, which is explained in terms of electron density. Namely, the average density along the centre of the [001] channel in GaAs is higher than in $Ga_{0.5}In_{0.5}P$, while it is the other way around along the off-centre-channel trajectory.

The effect of the interface between the layers of the lattice-matched multilayer solar cell, as well as the effect of strain, have been found to be negligible, which is understood, given the very similar chemistry and lattice constants of the three materials. The interface effect is found to be higher for lower velocity. At higher velocity, the energy loss can become negative, which effectively means that the proton loses less energy due to the interface.

Additional studies can help to understand the importance of the channelling effect in reducing the damage caused by radiation. The coupling between nuclear and electronic stopping power will be considered as a further step in this study.

Ethics. The authors declare that this manuscript conforms to the ethical code of the Royal Society.

Data accessibility. The data supporting the results presented in this article are available at the Dryad Digital Repository at the following link: https://doi.org/10.5061/dryad.rr4xgxd64. [73].

Authors' contributions. All authors contributed to the conception and design of the study, analysis and interpretation of results. Monte Carlo simulations were performed by F.D.P. and N.E.K. TDDFT calculations were performed by N.E.K. Initial draft was written by N.E.K. and F.D.P. E.A. provided a critical review of the manuscript. All authors reviewed the manuscript and gave final approval for publication.

Competing interests. We declare we have no competing interests.

Funding. The work presented has been funded by the Research Executive Agency under the European Union's Horizon 2020 Research and Innovation programme (project ESC2RAD: Enabling Smart Computations to study space RADiation effects, grant agreement no. 776410).

Acknowledgements. The authors thankfully acknowledge the computer resources at Donostia International Physics Center and at MareNostrum and the technical support provided by Barcelona Supercomputing Center (FI-2019-2-0017).

# Appendix A. SIESTA calculation parameters

All the DFT and RT-TDDFT calculations were performed using SIESTA version *siesta/4.1b4-trunk*-750. The plane-wave cut-off of the finite three-dimensional grid of 200 Ry was used.

The basis functions of the double-$\zeta$ polarized basis set have been generated as explained in [74,75]. The cut-off radii of the $\zeta_1$ function were defined by an energy shift of 272.1 meV (20 meV for $Ga_{0.5}In_{0.5}P$ and 10 meV in all the interface calculations), and the ones for the $\zeta_2$ function, using the split-norm parameter of 15% (30% for $Ga_{0.5}In_{0.5}P$ and in all the interface calculations). The parameters needed to generate the basis set in the calculation with $3d$-shell in Ge are listed in table 1. The extra $d$-shell is introduced as polarization orbital and has been obtained using the electric field polarization scheme described in [45], and, therefore, its finite-support radius coincides with that of the shell it polarizes. The double-$\zeta$ polarized basis set of the SIESTA code has previously been checked against plane-wave code ABINIT in similar calculations for Ge [32]. The agreement between the two codes for the band structure of Ge was excellent. Moreover, the validity of using the double-$\zeta$ polarized basis set in this work is verified by comparing our results for the stopping power of the $Ga_{0.5}In_{0.5}P$ with the plane-wave results from [31]. A reasonable agreement is observed for equivalent trajectories.

The pseudopotentials have been generated following [58] with the parameters listed in table 2. For all the elements except for H, the pseudopotentials were generated with the partial core correction [76], with the radii $r_{pc}$ given in table 2.

**Table 1.** Cut-off radii $r(\zeta_1)$ and $r(\zeta_2)$ of the first and second $\zeta$ functions, respectively (Bohr); soft-confinement potential prefactor $V_0$ (Ry); inner radius of the soft-confinement potential $r_i$.

| species | $n$ | $l$ | $r(\zeta_1)$ | $r(\zeta_2)$ | $V_0$ | $r_i/r(\zeta_1)$ |
|---------|-----|-----|--------------|--------------|-------|------------------|
| Ge      | 3   | 2   | 4.5          | 3.5          | 50    | 0.9              |
|         | 4   | 0   | 8.0          | 7.0          | 50    | 0.9              |
|         | 4   | 1   | 8.0          | 4.0          | 50    | 0.9              |
|         | 4   | 3   | 2.5          | 2.0          | 50    | 0.9              |

**Table 2.** Pseudopotentials cut-off radii for each angular-momentum channel (in Bohrs) and the partial core-correction radii $r_{pc}$ (Bohr).

| species | s | p | d | f | $r_{pc}$ |
|---|---|---|---|---|---|
| H ($1s^2$) | 1.25 | 1.25 | 1.25 | 1.25 | |
| Ge ($4s^24p^2$) | 2.25 | 2.99 | 2.48 | 2.48 | 1.958 |
| Ge ($3d^{10}4s^24p^2$) | 1.49 | 1.96 | 1.96 | 1.96 | 1.958 |
| Ga ($4s^24p^1$) | 2.18 | 2.35 | 2.53 | 2.53 | 1.389 |
| As ($4s^24p^3$) | 2.05 | 2.21 | 2.50 | 2.50 | 2.728 |
| In ($5s^25p^1$) | 2.51 | 2.61 | 2.85 | 2.42 | 1.395 |
| P ($3s^23p^3$) | 1.70 | 1.88 | 2.00 | 2.00 | 0.992 |

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
