## [Reviewer comments · Royal Society Open Science]

Review History

RSOS-200925.R0 (Original submission)

Review form: Reviewer 1

Is the manuscript scientifically sound in its present form?

Yes

Are the interpretations and conclusions justified by the results?

Yes

Is the language acceptable?

Yes

Do you have any ethical concerns with this paper?

No

Have you any concerns about statistical analyses in this paper?

No

Recommendation?

Accept as is

Comments to the Author(s)

Triple-junction solar cells have been widely used in aerospace. However, the solar cells can be damaged by the high-energy protons when they are exposed in the universe. This work, by using Monte Carlo simulation method, has comprehensively investigated how the protons impact the triple-junction GaInP/GaAs/Ge solar cells. This paper is very well prepared and the presented results are reliable. Therefore, I suggest accept as it is.

Review form: Reviewer 2

Is the manuscript scientifically sound in its present form?

Yes

Are the interpretations and conclusions justified by the results?

Yes

Is the language acceptable?

Yes

Do you have any ethical concerns with this paper?

No

Have you any concerns about statistical analyses in this paper?

No

Recommendation?

Accept with minor revision (please list in comments)

Comments to the Author(s)

The authors of the manuscript "Ab initio electronic stopping power for protons in Ga_{0.5}In_{0.5}P/GaAs/Ge triple-junction solar cells for space applications" use first-principles time-dependent density functional theory simulations and Monte-Carlo particle transport simulations to investigate proton penetration in these solar cells as a function of proton energy. The authors use state-of-the-art simulation techniques to study a timely question for an important material/device system and the results are timely. I recommend publication of this manuscript in Royal Society Open Science after the following points are addressed:

The authors use a double-zeta polarized numerical basis for their TDDFT simulations and it would be interesting to know whether this impacts the results?

The authors should also motivate how they chose their specific trajectories, in particular for the off-channeling cases. While it is interesting to see data for projectiles not moving on a channeling trajectory, it is not clear how important/relevant the specific trajectories chosen here are. What would be their weight if one were to sample all possible trajectories?

The color scheme of Fig. 3 is not a good choice and makes it difficult to discern the different data sets. It is also not clear why the thick purple line is so straight and does not exhibit any oscillations that can be attributed to atom positions?

At the bottom of page 10, the authors discuss the average electron density in SRIM vs. TDDFT. Can they quantify this somehow from their simulations?

On line 36 of page 11, the authors seem to imply that their off-channeling trajectories are not long enough for converged results? Can they either test this from their data and/or comment on what they mean by this and how much it is expected to impact the conclusions of this work?

I suggest to repeat the insets that illustrate the trajectories in Fig. 6, also for Fig. 8, if appropriate.

In addition, there is a typo on line 30 of page 6: "f or" should be "or" and on line 39 of page 11: "then" should be "than"

Decision letter (RSOS-200925.R0)

Dear Dr Koval

The Editors assigned to your paper RSOS-200925 "Ab initio electronic stopping power for protons in Ga_{0.5}In_{0.5}P/GaAs/Ge triple-junction solar cells for space applications" have now received comments from reviewers and would like you to revise the paper in accordance with the reviewer comments and any comments from the Editors. Please note this decision does not guarantee eventual acceptance.

Please submit your revised manuscript and required files (see below) no later than 21 days from today's (ie 08-Sep-2020) date. Note: the ScholarOne system will 'lock' if submission of the revision is attempted 21 or more days after the deadline. If you do not think you will be able to meet this deadline please contact the editorial office immediately.

on behalf of Professor Miles Padgett (Subject Editor)
openscience@royalsociety.org

Reviewer comments to Author:

Reviewer: 1

Comments to the Author(s)

Triple-junction solar cells have been widely used in aerospace. However, the solar cells can be damaged by the high-energy protons when they are exposed in the universe. This work, by using Monte Carlo simulation method, has comprehensively investigated how the protons impact the triple-junction GaInP/GaAs/Ge solar cells. This paper is very well prepared and the presented results are reliable. Therefore, I suggest accept as it is.

Reviewer: 2

Comments to the Author(s)

The authors of the manuscript "Ab initio electronic stopping power for protons in Ga_{0.5}In_{0.5}P/GaAs/Ge triple-junction solar cells for space applications" use first-principles time-dependent density functional theory simulations and Monte-Carlo particle transport simulations to investigate proton penetration in these solar cells as a function of proton energy. The authors use state-of-the-art simulation techniques to study a timely question for an important material/device system and the results are timely. I recommend publication of this manuscript in Royal Society Open Science after the following points are addressed:

The authors use a double-zeta polarized numerical basis for their TDDFT simulations and it would be interesting to know whether this impacts the results?

The authors should also motivate how they chose their specific trajectories, in particular for the off-channeling cases. While it is interesting to see data for projectiles not moving on a channeling trajectory, it is not clear how important/relevant the specific trajectories chosen here are. What would be their weight if one were to sample all possible trajectories?

The color scheme of Fig. 3 is not a good choice and makes it difficult to discern the different data sets. It is also not clear why the thick purple line is so straight and does not exhibit any oscillations that can be attributed to atom positions?

At the bottom of page 10, the authors discuss the average electron density in SRIM vs. TDDFT. Can they quantify this somehow from their simulations?

On line 36 of page 11, the authors seem to imply that their off-channeling trajectories are not long enough for converged results? Can they either test this from their data and/or comment on what they mean by this and how much it is expected to impact the conclusions of this work?

I suggest to repeat the insets that illustrate the trajectories in Fig. 6, also for Fig. 8, if appropriate.

In addition, there is a typo on line 30 of page 6: "f or" should be "or" and on line 39 of page 11: "then" should be "than"

===PREPARING YOUR MANUSCRIPT===

- one version identifying all the changes that have been

made (for instance, in coloured highlight, in bold text, or tracked changes);a 'clean' version of the new manuscript that incorporates the changes made, but does not highlight them. This version will be used for typesetting if your manuscript is accepted.
Please ensure that any equations included in the paper are editable text and not embedded images.

===PREPARING YOUR REVISION IN SCHOLARONE===

- If you are providing image files for potential cover images, please upload these at this step, and inform the editorial office you have done so. You must hold the copyright to any image provided.
- A copy of your point-by-point response to referees and Editors. This will expedite the preparation of your proof.

- Ensure that your data access statement meets the requirements at <https://royalsociety.org/journals/authors/author-guidelines/#data>. You should ensure that you cite the dataset in your reference list. If you have deposited data etc in the Dryad repository, please include both the 'For publication' link and 'For review' link at this stage.
- If you are requesting an article processing charge waiver, you must select the relevant waiver option (if requesting a discretionary waiver, the form should have been uploaded at Step 3 'File upload' above).
- If you have uploaded ESM files, please ensure you follow the guidance at <https://royalsociety.org/journals/authors/author-guidelines/#supplementary-material> to include a suitable title and informative caption. An example of appropriate titling and captioning may be found at https://figshare.com/articles/Table_S2_from_Is_there_a_trade-off_between_peak_performance_and_performance_breadth_across_temperatures_for_aerobic_scope_in_teleost_fishes_/3843624.

Author's Response to Decision Letter for (RSOS-200925.R0)

See Appendix A.

RSOS-200925.R1 (Revision)

Review form: Reviewer 2

Is the manuscript scientifically sound in its present form?

Yes

Are the interpretations and conclusions justified by the results?

Yes

Is the language acceptable?

Yes

Do you have any ethical concerns with this paper?

No

Have you any concerns about statistical analyses in this paper?

No

Recommendation?

Accept as is

Comments to the Author(s)

I recommend the revised manuscript for publication.

Decision letter (RSOS-200925.R1)

Dear Dr Koval,

It is a pleasure to accept your manuscript entitled "Ab initio electronic stopping power for protons in Ga_{0.5}In_{0.5}P/GaAs/Ge triple-junction solar cells for space applications" in its current form for publication in Royal Society Open Science. The comments of the reviewer(s) who reviewed your manuscript are included at the foot of this letter.

on behalf of Professor Miles Padgett (Subject Editor)
openscience@royalsociety.org

Reviewer comments to Author:
Reviewer: 2
Comments to the Author(s)

I recommend the revised manuscript for publication.

Appendix A

Response to the Reviewers reports for the manuscript RSOS-200925

We would like to thank the Reviewers for their comments and for their assessment of our results as reliable and timely. We have addressed all of the questions and comments raised by the Reviewers and modified the manuscript accordingly.

List of changes (page numbers are those printed in black squares):

1. On page 5, corrected typo “f or” to “for”.
2. On page 5, 5th paragraph, modified:
old: *but barely noticeable for the mid-channel trajectory.*
new: *but are not noticeable for the mid-channel trajectory, along which the density is quite homogeneous.*
3. Figure 3: changed colors and lines types (page 6) and modified the caption.
4. On page 9(bottom)-10, added a sentence:
The average valence density in Ge is about $0.179 \text{ electrons}/\text{\AA}^3$, which corresponds to the trajectory 3 in figure 7c, while the average density along the trajectories 1 and 2 is slightly lower.
5. On page 10, first paragraph, modified:
old: *In figure 6a, we show as well the ESP for an off-channel proton trajectory which makes 14° with the direction $[001]$ (z-axis).*
new: *In figure 6a, we show as well the ESP for an off-channel proton trajectory (random direction, not parallel to any channel) in order to see how well it compares to SRIM. The random trajectory is arbitrarily chosen so that it makes 14° with the direction $[001]$ (z-axis).*
6. On page 10, first paragraph, modified the sentence:
old: *Possible reasons for that is the absence of core electrons, as discussed in the next subsection, and insufficient trajectory sampling when calculating the ESP for the off-channelling case (i.e., a longer propagation is probably needed).*
new: *It is known that longer trajectories would affect our results, but on a smaller scale than the observed discrepancy [32]. A possible reason for that is the absence of core electrons, as discussed in the next subsection. However, SRIM is expected to be inaccurate on that scale as well [72], and further expense in calculations is therefore not justified.*
7. On page 10, corrected typo “then” to “than”.
8. Figure 8: added insets with schemes of the proton positions and modified the caption.
9. Appendix A, page 16, modified first sentence of the second paragraph:

old: *The basis functions have been generated as explained in Refs. [74,75].*

new: *The basis functions of the double- ζ polarized basis set have been generated as explained in Refs. [74,75].*

10. Appendix A, pages 16-17, added at the end of the second paragraph:
The double- ζ polarized basis set of the SIESTA code have previously been checked against plane-wave code ABINIT in similar calculations for Ge [33]. The agreement between the two codes for the band structure of Ge was excellent. Moreover, the validity of using the double- ζ polarized basis set in this work is verified by comparing our results for the stopping power of the Ga_{0.5}In_{0.5}P with the plane-wave results from Ref. [32]. A reasonable agreement is observed for equivalent trajectories.

11. Data Accessibility statement (page 17): added DOI for the data uploaded to the Dryad Repository and the corresponding citation to the reference list.

Reviewer: 1

Comments to the Author(s)

Triple-junction solar cells have been widely used in aerospace. However, the solar cells can be damaged by the high-energy protons when they are exposed in the universe. This work, by using Monte Carlo simulation method, has comprehensively investigated how the protons impact the triple-junction GaInP/GaAs/Ge solar cells. This paper is very well prepared and the presented results are reliable. Therefore, I suggest accept as it is.

Authors: We thank the Reviewer for the positive evaluation of our work and for recommending our manuscript for publication.

Reviewer: 2

Comments to the Author(s)

The authors of the manuscript "Ab initio electronic stopping power for protons in Ga_{0.5}In_{0.5}P/GaAs/Ge triple-junction solar cells for space applications" use first-principles time-dependent density functional theory simulations and Monte-Carlo particle transport simulations to investigate proton penetration in these solar cells as a function of proton energy. The authors use state-of-the-art simulation techniques to study a timely question for an important material/device system and the results are timely. I recommend publication of this manuscript in Royal Society Open Science after the following points are addressed:

Authors: First of all, we thank the Reviewer for recommending publication of our manuscript. We address all the questions/suggestions raised by the Reviewer below.

1. The authors use a double-zeta polarized numerical basis for their TDDFT

simulations and it would be interesting to know whether this impacts the results?

Authors: The double-zeta polarized basis set of the SIESTA code have previously been checked against plane wave code ABINIT in similar calculations for Ge (please see Ullah et al, Phys. Rev. B 91, 125203 (2015)). The agreement between the two codes for the band structure of Ge was excellent. Moreover, in this work, we compare our results for the stopping power of the Ga_{0.5}In_{0.5}P with the plane wave results from Lee et al, Eur. Phys. J. B 91(10), 222, and also observe very good agreement for equivalent trajectories. This information has been added to Appendix A, we thank the Reviewer for pointing us to that omission.

2. The authors should also motivate how they chose their specific trajectories, in particular for the off-channeling cases. While it is interesting to see data for projectiles not moving on a channeling trajectory, it is not clear how important/relevant the specific trajectories chosen here are. What would be their weight if one were to sample all possible trajectories?

Authors: The purpose of our work is to focus on the channeling effect, thus most of the trajectories are arbitrarily chosen along the channels (closer or farther away from the target atoms). Two crystallographic directions are considered, [001] and [011], because these are very different channels both in structure and density and therefore are interesting to compare. The third main direction [111] is not taken into account, since we already know from the previous work on Ge that the stopping power in this channel is very similar to the channel along the [001] direction (Ullah et al, Phys. Rev. B 91, 125203 (2015), Fig. 3).

The sampling technique would give similar results to the case of the off-channeling (random) trajectory which gives a good estimate of the experimental (SRIM) stopping power (see Schleife et al, Phys. Rev. B 91, 014306 (2015)). We show in our manuscript, for comparison purpose, one result for the off-channeling trajectory, and also results from Lee et al, Eur. Phys. J. B 91(10), 222. However, the main aim of our work is different, i.e., it is rather to show how different the stopping power can be depending on whether the proton moves far away or close to the target atoms inside the channels, thus emphasizing the channeling effect.

We slightly modified the corresponding text (page 10, first paragraph):

In figure 6a, we show as well the ESP for an off-channel proton trajectory (**random direction, not parallel to any channel**) in order to see how well it compares to SRIM. **The random trajectory is arbitrarily chosen so that** it makes 14° with the direction [001] (z-axis).

3. The color scheme of Fig. 3 is not a good choice and makes it difficult to discern the different data sets. It is also not clear why the thick purple line is so straight and does not exhibit any oscillations that can be attributed to atom positions?

Authors: We changed the color scheme of Fig.3 and hope that it is now easier to analyze. We slightly modified the caption of the figure as well:

Figure 3. Position-dependent total electronic energy for two trajectories of the proton moving through Ge with velocity $v = 0.5$ a.u. along the [011] direction. The energy is shown for the proton moving along the center of the channel (impact parameter of 1.98 \AA , red long-dashed line) and along the off-center channeling trajectory (impact parameter of 0.7 \AA , blue line). The linear fitting is shown for each case.

Regarding the straight line, corresponding to the center of the [011] channel (the trajectory most distant from all the atoms), there are no perceptible oscillations because the density is quite homogeneous along the center of the channel. We modified the corresponding sentence from

“but barely noticeable for the mid-channel trajectory.”

to:

“but are not noticeable for the mid-channel trajectory, along which the density is quite homogeneous.”

4. At the bottom of page 10, the authors discuss the average electron density in SRIM vs. TDDFT. Can they quantify this somehow from their simulations?

Authors: We believe the Reviewer refers to the phrase “This suggests that in the SRIM calculations, the average density is higher than the density on most trajectories in the RT-TDDFT calculations for Ge.” In our SRIM calculations we used the (atomic) density corresponding to the lattice constant used in RT-TDDFT calculations for each material. However, in SRIM the density is homogeneous throughout the system, while in our calculations, the density is lower than average in the middle of the channels. Regarding the electron density, the average valence electron density in our RT-TDDFT calculations for Ge is ~ 0.179 electrons/ \AA^3 . This value corresponds to the average density along the trajectory 3 in figure 7(c). On trajectories 1 and 2, the density is slightly lower. We added this information to the manuscript at the bottom of page 9-top of page 10 (pp. 10-11 in the line-numbered proof).

5. On line 36 of page 11, the authors seem to imply that their off-channeling trajectories are not long enough for converged results? Can they either test this from their data and/or comment on what they mean by this and how much it is expected to impact the conclusions of this work?

Authors: We mention that probably longer propagation would improve the comparison to SRIM. However, similar discrepancies are observed in other works, which perform longer propagation and use plane wave basis set (one example is again the work we cite - Lee et al, Eur. Phys. J. B 91(10), 222). We think that there are many reasons for the slight discrepancy between *ab initio* results and SRIM, the unit cell size and the absence of core electrons are some of them. Nevertheless, larger unit cell and use of

core electrons would only slightly improve the results, while being much more computationally expensive.

We modified the corresponding part of the text:

The text now: The resulting ESP for the off-channel trajectory is in between the values for two off-center channeling trajectories 2 and 3, and thus does not bring the channeling results closer to SRIM. Possible reasons for that is the absence of core electrons, as discussed in the next subsection, and insufficient trajectory sampling when calculating the ESP for the off-channelling case (i.e., a longer propagation is probably needed). Other reasons may come from the inaccuracy of SRIM.

New text: The resulting ESP for the off-channel trajectory is in between the values for two off-center channeling trajectories 2 and 3, and thus does not bring the channeling results closer to SRIM. **It is known that longer trajectories would affect our results, but on a smaller scale than the observed discrepancy [32]. A possible reason for that is the absence of core electrons, as discussed in the next subsection. However, SRIM is expected to be inaccurate on that scale as well [72], and further expense in calculations is therefore not justified.**

6. I suggest to repeat the insets that illustrate the trajectories in Fig. 6, also for Fig. 8, if appropriate.

Authors: We added insets to each panel of figure 8 and modified accordingly the caption of the figure, adding:

The inset on each panel shows a schematic view of the channel section and the proton position according to the impact parameter.

7. In addition, there is a typo on line 30 of page 6: "f or" should be "or" and on line 39 of page 11: "then" should be "than"

Authors: We thank the Reviewer for noticing that. We corrected the typos.

Sincerely,

Natalia E. Koval, Fabiana Da Pieve, and Emilio Artacho